# Cuticular Hydrocarbon Recognition in the Mating Behavior of Two *Pissodes* Species

**DOI:** 10.3390/insects10070217

**Published:** 2019-07-23

**Authors:** Ruixu Chen, Tian Xu, Dejun Hao, Stephen A. Teale

**Affiliations:** 1Co-Innovation Center for the Sustainable Forestry in Southern China, Nanjing Forestry University, Nanjing 210037, China; 2College of Forestry, Nanjing Forestry University, Nanjing 210037, China; 3College of Environmental Science and Forestry, State University of New York, Syracuse, NY 13210, USA

**Keywords:** cuticular hydrocarbon, copulation behavior, *Pissodes strobi*, *Pissodes nemorensis*

## Abstract

Two sibling weevil species, *Pissodes strobi* Peck and *P. nemorensis* Germar (Coleoptera: Curculionidae), can form reduced-fitness hybrids in the laboratory, but neither their premating isolation mechanisms nor mating behaviors are well-understood. Cuticular hydrocarbons (CHCs) have been reported as crucial chemical cues in mating recognition in many insects, including weevils, and, thus, may also mediate the mating behavior of *P. strobi* and *P. nemorensis*. We conducted a series of behavioral observations, bioassays, and chemical analyses to investigate the role of CHCs in their mating behavior. Copulation behavior of both species followed similar steps: approaching, mounting, tapping, aedeagus extrusion, and copulation. In *P. strobi*, hexane extraction significantly reduced the number of successful male copulations compared with freeze-killed females. Conversely, significantly fewer *P. nemorensis* males copulated with dead females compared with live females. No significant differences were detected among hexane-extracted, freeze-killed or recoated female carcasses to *P. nemorensis*. These findings suggested that female cuticular extracts contain important cues in mate recognition in *P. strobi* but not in *P. nemorensis*. We identified 21 CHCs from both species with variation in abundances between sexes and seasons. Discriminant analysis revealed incomplete overlap of CHC compositions in females of the two species in summer, when hybridization potentially occurs.

## 1. Introduction

Cuticular hydrocarbons (CHCs) are nonvolatile long-chain hydrocarbons that protect insects from desiccation, pathogens, and mechanical damage [1]. They also have an important role in insect chemical communication, such as mate recognition and stimulating copulation [2]. Extensive studies have revealed that CHCs act as contact sex pheromones in the mating of some coleopteran species, such as Coccinellidae [3], Curculionidae [4], and Cerambycidae [5,6,7]. In weevils, the behaviors triggered by contact sex pheromones are similar across species, with only a few differences, mainly related to precopulation, copulation, and postcopulation [4,8,9,10,11,12,13,14,15,16,17,18].

*Pissodes strobi* Peck and *Pissodes nemorensis* Germar (Coleoptera: Curculionidae) are sibling species that damage a range of *Pinus* and *Picea* species in North America with one generation per year [19,20,21]. These two weevil species are morphologically similar and can mate, resulting in hybrid offspring [21,22,23]. In early spring, adult weevils of both species emerge from their overwintering sites in the forest floor, feed, mate and lay eggs on their own habits [19,24]. *Pissodes strobi* emerge, then feed and oviposit on leaders, whereas *P. nemorensis* feed and oviposit on slash and the lower stems of healthy and unhealthy trees [25]. However, during the late summer and fall, the newly emerged adults of both species feed on the lateral branches and shoots of the trees. When temperature drop and photoperiod shorten, the weevils go into hibernation in the duff [26].

Grandisol (cis-2-isopropenyl-1-methylcyclobutaneethanol) and its corresponding aldehyde, grandisal have been isolated from male *P. nemorensis* in which they function as aggregation pheromone components [27]. *P. strobi* males also produce grandisol and grandisal but neither males nor females have been shown to be attracted to them [27,28]. Interestingly, the presence of male *P. strobi* inhibits attraction of *P. nemorensis* males and females to *P. nemorensis* males or synthetic *P. nemorensis* pheromone [21]. Although the enantiomeric composition of both compounds produced by these species is different, the behavioral role of stereochemistry in interspecific interactions remains unclear [27]. Furthermore, the CHC composition and function in the mating behavior of *P. strobi* and *P. nemorensis* are unclear.

Here, we report on a series of behavioral observations, bioassays, and chemical analyses we conducted to test the following hypotheses based on the life history and chemical ecology of *P. strobi* and *P. nemorensis*: (1) both species use CHCs as contact sex pheromones; (2) in early spring, there are interspecific differences in cuticular hydrocarbon composition for the two species; (3) in late summer and fall, there are no cuticular hydrocarbon composition differences for the two species; (4) the total cuticular hydrocarbon quantity is greater in *P. strobi* in the spring due to the desiccating conditions of wind and sun exposure on pine leaders compared to that of *P. nemorensis* whose spring habitat is beneath the forest canopy; (5) The total cuticular hydrocarbon quantity is the same on these two species in late summer when they both occur on pine laterals.

## 2. Materials and Methods

### 2.1. Insect Sources

Spring populations of *P. strobi* and *P. nemorensis* were collected in April 2018. *P. strobi* adults were collected from white pine, *Pinus strobus*, leaders near Star Lake (44.167551, −75.048181), NY, USA, and *P. nemorensis* adults were collected from fallen red pine, *Pinus resinosa*, in the Svend O. Heiberg Memorial forest in Tully (42.773047, −76.091273), NY, USA. Infested host material was collected from the same place in August 2018 at the same locations and transported to the laboratory in Syracuse before adult eclosion. Infested white pine leaders were kept in a metal can fitted with a glass jar for *P. strobi* adult collection. Infested red pine slash was waxed and kept in emergence chambers with a glass jar for collecting emerged *P. nemorensis* adults. Emerged adults were placed in 3.8 L glass jars in an environment chamber under a 16 h: 8 h light/dark photoperiod, 60% humidity and multiple temperature program: 23 °C (6:30 a.m.–8:30 a.m.), 25 °C (8:30 a.m.–10:30 a.m.), 28 °C (10:30 a.m.–14:30 p.m.), 25 °C (14:30 p.m.–18:30 p.m.), 23 °C (18:30 p.m.–23:30 p.m.) and finally 20 °C (23:30 p.m.–6:30 a.m.). The eclosed adults were sexed following the method of Harman and Kuman [29]. Males and females were kept separately in clear glass jars with fresh white pine branches as food until further use.

### 2.2. Mating Recognition Bioassays

The summer populations of weevils of both species were used in mating behavior bioassays that followed Ginzel and Hanks [6] with slight modification. Males of each species were tested with the following treatments: (1) live conspecific females (controls); (2) freeze-killed conspecific female carcasses; (3) hexane-extracted conspecific female carcasses; (4) conspecific female carcasses extracted then recoated with 0.05 female-equivalent (FE) conspecific CHC extract; (5) live heterospecific females; (6) freeze-killed heterospecific female carcasses; (7) hexane-extracted heterospecific female carcasses; and (8) heterospecific female carcasses recoated with 0.05 FE conspecific CHC extract. All treatments were replicated 20 times. In (1) and (5), a pair of live adults was placed in a Petri dish (10 cm diameter) and observed for 1 h. In (2) and (6), one freeze-killed female carcass (−20 °C for 1 h) was warmed for 30 min under room temperature and then presented to a live male, as described earlier. In (3) and (7), the freeze-killed female carcass was rinsed with hexane for 20 s to remove CHCs and dried for 30 min to allow the solvent to evaporate. The carcass was then presented to a live male, as described earlier. In (4) and (8), the same female carcass was recoated with 0.05 FE and dried for 30 min to allow the solvent to evaporate. The carcass was then presented to a live male, as described earlier. A successful copulation was recorded if the male inserted his aedeagus into the female genitalia.

### 2.3. Chemical Analysis of Whole-Body Extraction

For both species and sexes, five adults were selected from the spring and summer populations and then starved for 24 h. Weevils were kept individually in vials and then freeze-killed (−20 °C for 1 h). Each weevil was immersed in 400 μL hexane in an Agilent amber vial (2 mL) for 20 s with light agitation. The carcasses were then transferred to a vial and stored in a freezer. Extraction weevils were evaporated to dryness under a stream of nitrogen. The CHCs were then re-dissolved in 100 μL hexane (OmniSolv, Darmstadt, Germany, Chromatography Grade). Tricosane (Sigma-Aldrich, St. Louis, MO, USA; chemical purity 99%) was added to each sample as an internal standard for quantification. Extractions were analyzed by coupled gas chromatography-mass spectrometry (GC-MS; Agilent 7890A GC interfaced to a 5975-mass selective detector in EI mode, 70 eV; Agilent Technologies, Santa Clara, CA, USA), fitted with a DB5-MS capillary column (30 m × 0.25 mm ID × 0.25 μm film thickness; Agilent Technologies). A 1-μL sample was injected in splitless mode with helium as the carrier gas (1 mL/min). The GC temperature program was 60~°C (1 min) to 200°C at 20 °C/min and then increased to 315 °C at 5 °C/min. The MS scan rang was 33–650 *m*/*z*. The final temperature was held for 15 min. 500 ng/mL C7–C40 alkane standards (Sigma-Aldrich, chemical purity 99%) diluted in hexane were injected with the same program. The CHC components were identified by comparing their linear retention indices and mass spectra with the standard spectra in the library database (NIST2.0) and NIST webbook (http://webbook.nist.gov/chemistry) [30].

For both species and sexes, five adults were selected from the spring and summer populations and then starved for 24 h. Weevils were kept individually in vials and then freeze-killed (−20 °C for 1 h). Each weevil was immersed in 400 μL hexane in an Agilent amber vial (2 mL) for 20 s with light agitation. The carcasses were then transferred to a vial and stored in a freezer. Extraction weevils were evaporated to dryness under a stream of nitrogen. The CHCs were then re-dissolved in 100 μL hexane (OmniSolv, Darmstadt, Germany, Chromatography Grade). Tricosane (Sigma-Aldrich, St. Louis, MO, USA; chemical purity 99%) was added to each sample as an internal standard for quantification. Extractions were analyzed by coupled gas chromatography-mass spectrometry (GC-MS; Agilent 7890A GC interfaced to a 5975-mass selective detector in EI mode, 70 eV; Agilent Technologies, Santa Clara, CA, USA), fitted with a DB5-MS capillary column (30 m × 0.25 mm ID × 0.25 μm film thickness; Agilent Technologies). A 1-μL sample was injected in splitless mode with helium as the carrier gas (1 mL/min). The GC temperature program was 60~°C (1 min) to 200°C at 20 °C/min and then increased to 315 °C at 5 °C/min. The MS scan rang was 33–650 *m*/*z*. The final temperature was held for 15 min. 500 ng/mL C7–C40 alkane standards (Sigma-Aldrich, chemical purity 99%) diluted in hexane were injected with the same program. The CHC components were identified by comparing their linear retention indices and mass spectra with the standard spectra in the library database (NIST2.0) and NIST webbook (http://webbook.nist.gov/chemistry) [30].

### 2.4. Statistical Analyses

Statistical analyses were performed using Microsoft Excel (Microsoft office 365. Released 2017. Redmond, WA, USA) and SPSS 21 (IBM Corp. Released 2012. IBM SPSS Statistics for Windows, Version 21.0. Armonk, NY, USA). The bioassay results were formed in crosstables and analyzed by χ^2^ or Fisher exact tests when the expected frequencies were <5. The concentration of CHCs in each weevil was calculated using an internal standard. Data were first checked for normality using the Shapiro–Wilk test and then transformed if they were non-normally distributed. Levene’s test was used for homogeneity of variance. Quantitative differences in individual CHC components between the two species, between the sexes of both species, and among the seasonal populations were then compared by using independent *t*-tests. Discriminant analysis (DA) was used to explore the relationship of the relative abundances of the identified CHCs among species, sexes, and populations (using Bayes discriminant function and Canonical discriminant function). A Kruskal–Wallis one-way ANOVA was used to compare the relative abundances of the CHCs in the extractions from females of both species from the spring and summer populations.

## 3. Results

### 3.1. Mating Behavior

The mating behavior could be separated into five steps: (1) approaching, in which the male encountered the female (carcass) followed by rapid contact with its antennae and rostrum; (2) mounting, in which the male mounted the female (carcass); (3) tapping, in which the male touched the head and tergum of the female with its rostrum, antennae, and forelegs. Some males occasionally left the females at this point; (4) aedeagus extrusion, in which the male first adjusted its body to face the same direction as the female (carcass) and then moved back until its aedeagus could reach the female genitalia. The male extruded its aedeagus and touched the female abdomen with it a few times before (5) copulating, in which the male inserted its aedeagus into the bursa copulatrix. The duration of a single copulation varied from 2 min to 1 h (end of observation period). Males also demonstrated postcopulatory guarding behavior.

### 3.2. Function of Cuticular Hydrocarbons in Mating Behavior

In *P. strobi*, significantly more males copulated with live females (control) than with the hexane-extracted female carcasses (*p* < 0.001) and recoated carcass (*p* = 0.047). A significant difference was also detected between the numbers of males that copulated with freeze-killed and hexane-extracted female carcasses (*p* = 0.003), but not between the controls and freeze-killed carcasses (*p* = 0.525) (Figure 1a); In *P. nemorensis*, significantly more males copulated with the controls than with either the freeze-killed (*p* = 0.028) or hexane-extracted carcasses (*p* = 0.020), whereas the numbers of males that copulated with the controls and the recoated carcasses showed no significant difference (*p* = 0.077) (Figure 1a).

Significantly more male *P. strobi* copulated with live female *P. nemorensis* females (controls) than with freeze-killed carcasses (*p* = 0.027), the hexane-extracted carcasses (*p* < 0.001) and the recoated carcasses (*p* = 0.001) (Figure 1b). However, no significant difference was detected between the freeze-killed and hexane-extracted carcasses (*p* = 0.235) or between freeze-killed and recoated carcasses (*p* = 0.451). Male *P. nemorensis* only responded to the live female *P. strobi* and the corresponding freeze-killed carcasses, but with no significant difference (*p* = 0.077) (Figure 1b).

When the spring and summer samples were analyzed by discriminant analysis separately, they were both classified into four groups (Figure 2a,b) by function 1 and 2 that showed significant difference in Wilk’s Lambda test. In the summer population, male *P. nemorensis* were separated by function 1 (92.8%) from female *P. nemorensis* and both sexes of *P. strobi*, whereas the females of *P. strobi* and *P. nemorensis* shared a similar CHC profile (Figure 2a). In the spring population, the males and females of the two species were well separated by function 1 (85.5%) (Figure 2b).

### 3.3. Chemical Analyses of Cuticular Hydrocarbons

In both the spring and summer populations, the CHCs of male and female *P. strobi* and *P. nemorensis* shared a similar profile but with varied relative abundances (Appendix A). In total, 21 CHC components were extracted from female *P. strobi* and *P. nemorensis* (Figure 3), including n-alkanes (5/21), mono-methylalkanes (11/21), a di-methylalkane (1/21), monoenes (2/21), and dienes (2/21). n-Alkanes ranged from C25 to C29, with odd numbers of carbons predominating. The dominant n-alkane was C27 in spring and C29 in summer for both species. Methylalkanes were the most abundant components of the CHCs, with percentages ranging from 48.43% to 62.45% among both populations. 5-Methyltritriacontane was the most abundant component, followed by nonacosane and heptacosane. 11,15-Dimethylheptacosane and 5-methylnonacosane were only found in spring populations. Seasonal differences were statistically significant among the relative abundances of pentacosane, 3-methylheptacosane, 5-methylnonacosane, and tritriacontadiene in female *P. strobi* (Figure 3).

Statistical analyses revealed significant differences in the average amounts of CHCs (Table 1) on both sexes of *P. strobi* (*t* = −3.375, *df* = 8, *p* = 0.010 for males; *t* = −2.627, *df* = 5.571, *p* = 0.042 for females) between spring and summer populations, whereas no significant difference was found in the average CHC amounts of both sexes of *P. nemorensis* between the two seasonal populations. Male *P. strobi* had significantly larger CHC amounts than male *P. nemorensis* in both spring (*t* = −2.659, *df* = 8, *p* = 0.029) and summer (*t* = −2.676, *df* = 8, *p* = 0.028). Female *P. strobi* showed significantly larger CHC amounts than female *P. nemorensis* in spring (*t* = −2.446, *df* = 8, *p* = 0.040), but not in summer (*t* = −0.746, *df* = 8, *p* = 0.477).

## 4. Discussion

Mating behavior is a critically important component of insect behavioral ecology [31]. The mating behavior patterns we observed in *P. strobi* and *P. nemorensis* were similar to those observed in other curculionids. Tapping (Step 3) in the mating behavior of weevils was reported to prevent females from struggling and to prepare them for copulation [9,16,18]. Our bioassay results showed that most of the male *P. strobi* and *P. nemorensis* recognized either live or freeze-killed females via rapid contact with the female body surface on their first encounter. Only a few males left the female after mounting and tapping, suggesting that some unknown cues, possibly chemicals, on the female body surface trigger male mating behavior and that behavioral responses from females are not required at this step. Female CHCs have been shown to regulate mating behavior at a short distance in the raspberry weevil [4]. The tapping behavior performed by male *P. strobi* and *P. nemorensis* after mounting could be further confirmation for sexual recognition or stimulation for the behaviors that then follow.

Male *P. strobi* and *P. nemorensis* responded differently in the con- and heterospecific bioassays, revealing differences in their mate recognition systems. In the conspecific tests, no male *P. strobi* mated with the hexane-extracted female carcasses, whereas some males mated with the recoated carcasses. This suggests that CHCs are key cues for recognizing mates in *P. strobi*. By contrast, in *P. nemorensis*, the number of males responding to freeze-killed females was significantly reduced compared with live females, whereas solvent extraction only slightly decreased the male response compared with fresh carcasses, suggesting that the chemicals extracted from the female were insufficient for mate recognition.

The heterospecific tests indicated that CHCs also mediate interspecific interactions for *P. nemorensis* males because solvent extraction significantly reduced the number of successful copulations. Recoating extracted *P. strobi* females did not result in an increase in the number of male *P. nemorensis* that responded, possibly for the same reason as discussed earlier. Compared with bioassay results in other weevil species, such as the raspberry weevil [4], the ratio of responding *Pissodes* males in our tests were lower, which might be because: (1) the recoated CHC concentration (0.05 FE) was insufficient for mate recognition compared with freeze-killed carcasses or live females; and (2) the trials were conducted in Petri dishes (10 cm diameter), larger than those used by Mutis et al. (5 cm diameter) [4], which might reduce the frequency of chance encounters.

The relative abundance of the CHC components in our samples was further investigated by DA. The CHC compositions of both sexes of both species were distinct in the spring (Figure 2). However, in summer, when breeding site specificity may no longer prevent interspecific contact [24], the CHC profiles of the females of both species did not separate on the first discriminant function, whereas those of the males did. The similarity between the female CHCs could fail to prevent hybridization between these species during the summer.

Similar to that reported for other adult weevils, saturated hydrocarbons were the most abundant components among the surface lipids, with odd-numbered alkanes such as C27 and C29 predominating, indicating the formation from two carbon units followed by decarboxylation [4,32,33,34] (Appendix A). Small amounts of even-numbered carbon chains were also present, possibly originating from chain initiation with a propionyl-CoA, instead of an acetyl-CoA [35]. In the current study, compound quantities varied between spring and summer populations and these seasonal differences were more significant than the differences between sexes or species (Appendix A). No sex-specific compounds were found for either species, but the quantities of CHCs differed among the sexes and seasonal populations. Therfore, we hypothesize that mate recognition in *P. strobi* and *P. nemorensis* is mediated by a blend of several compounds. For example, Sun [36] reported that n-pentacosane could elicit courtship behavior in the male tea weevil, *Myllocerinus aurolineatus* (Coleoptera: Curculionidae)*,* but less so than the crude extract. Therefore, n-pentacosane might not be the sole component of the contact sex pheromone of *M. aurolineatus*. Some unsaturated alkenes were minor components in our extractions. Alkenes have seldom been reported in weevils and their functions are largely unknown [32,37]. However, they have been shown to be important components in insect chemical communication. For example, the first cuticular alkene shown to be a pheromone was (Z)-9-tricosene, isolated from the house fly, *Musca domestica* (Diptera: Muscidae) [38]. Some of the alkenes in our samples showed quantitative differences between sexes or species, suggesting that they may function in mate recognition.

Total quantities of CHCs in both sexes of *P. strobi* and *P. nemorensis* were also calculated for two seasonal populations. Previous studies suggested that CHCs are effective in waterproofing the insect surface [39,40]. Monzer [34] reported that CHCs on red palm weevils helped these insects to tolerate desiccation. In the current study, the greater amount of CHCs likely protects the spring *P. strobi* adults from evaporative water loss due to sun exposure and wind while they are on leaders. By contrast, during the summer, when these species inhabit the same environment, they showed similar quantities of CHC, which might also contribute to hybridization between them during the summer.

## 5. Conclusions

In this study, we provide the first evidence for the existence of contact pheromone, probably formulated by CHCs, that mediates copulation behavior and hybridization in *P. strobi* and *P. nemorensis*. Our findings suggest that distinct differences in CHCs in these species may contribute to the prevention of hybridization between these forest pests. Meanwhile, the CHC composition might be both influenced by food sources and environmental factors.

**Note:** This study was conducted in Stephen A. Teale’s lab, in College of Environmental Science and Forestry, State University of New York.

## Figures and Tables

**Figure 1 insects-10-00217-f001:**
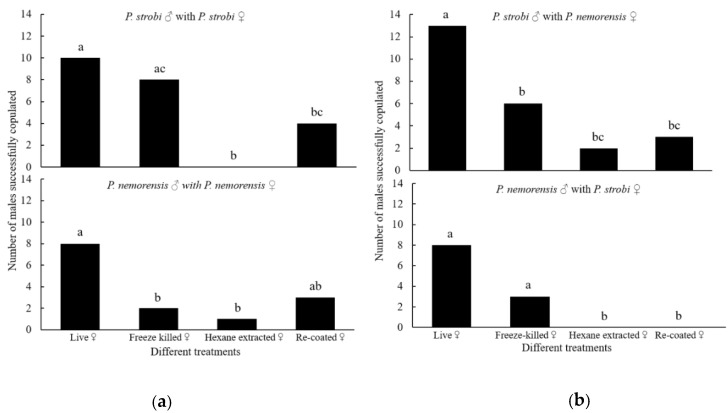
The numbers of male *Pissodes strobi* and *Pissodes nemorensis* that copulated with intraspecific (**a**) and interspecific (**b**) live females, freeze-killed females, hexane-extracted female carcasses, and recoated female carcasses. For each test, n = 20. Different lowercase letters above the bars indicate significant differences among treatments at α = 0.05 by χ^2^ or Fisher exact tests.

**Figure 2 insects-10-00217-f002:**
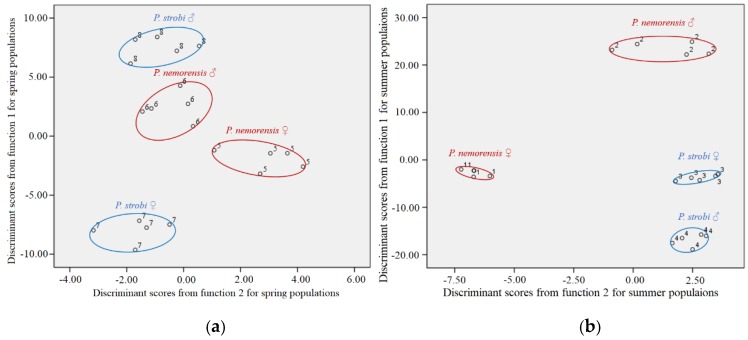
Scatter plots from discriminant analyses (DA) for the cuticular hydrocarbon (CHC) profiles of both sexes of *Pissodes strobi* and *Pissodes nemorensis* in the spring population (**a**), summer population (**b**). 1, 2, 3 and 4 represent female *P. nemorensis*, male *P. nemorensis*, female *P. strobi*, and male *P. strobi* in summer, respectively; 5, 6, 7, and 8 represent female *P. nemorensis*, male *P. nemorensis*, female *P. strobi* and male *P. strobi* in spring, respectively.

**Figure 3 insects-10-00217-f003:**
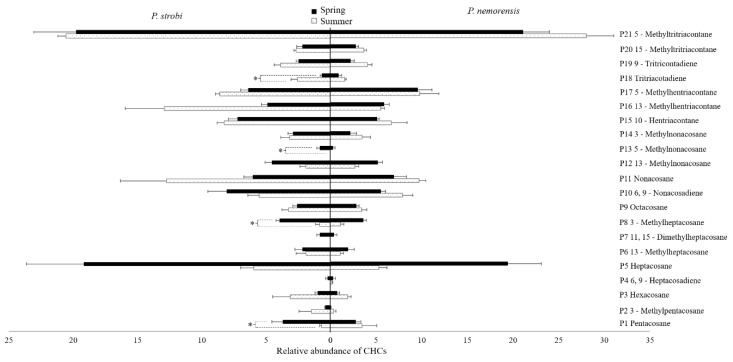
Relative abundances of the CHCs on the body surface of female *Pissodes strobi* and *Pissodes nemorensis* females in spring and summer populations. Mean ± SE; n = 5. The bars in black and white represent the relative abundances of the CHCs extracted from the females in spring and summer populations, correspondingly. * Indicates a significant difference in the relative abundances of each component between spring and summer females at α = 0.05.

**Table 1 insects-10-00217-t001:** The average amounts of CHCs per weevil (μg) of both sexes from the spring and summer populations of *P. strobi* and *P. nemorensis*
^†,‡^.

Population	*P. strobi*		*P. nemorensis*	
	Male	Female	Male	Female
**Spring**	2.325 ± 0.372 ^a,A^	2.289 ± 0.435 ^a,A^	1.084 ± 0.282 ^B^	1.010 ± 0.290 ^B^
**Summer**	1.252 ± 0.168 ^b,A^	0.812 ± 0.068 ^b,A^	0.719 ± 0.106 ^B^	0.729 ± 0.089 ^A^

^†^ Mean ± SE; n = 5; statistical differences were determined by independent *t*-tests at α = 0.05; ^‡^ Different lowercase letters indicate significant differences between seasons (in the same row) and uppercase letters indicate significant differences applies to species comparison for sexes separately.

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
