# Peer review of "Cuticular Hydrocarbon Recognition in the Mating Behavior of Two Pissodes Species"

_insects, 2019, doi:10.3390/insects10070217_

Round 1
Reviewer 1 Report
This is a solid and interesting analysis of the CHCs of two closely related weevil species. I missed some information on the life cycles of the species and have some suggestions for more reader-friendly presentation.
It appears (l. 69-73) that the spring and the summer populations of the same year are two different generations, but we are not informed whether they have one or two generations per year. The question is relevant because the mating tests were made only with the summer population. If there is only one generation that breeds in spring (which is the typical life cycle where I come from), then the animals seem to mate both in summer and again (after hibernation) in spring, but the spring matings are probably the most important. In this sense I can understand why the authors expect similar CHC composition and quantity at the two seasons. If there are two generations per year, this is not obvious. Please clear this up, and explain why only the summer animals were used for the mating test.
Some formalistic suggestions:
Please add the family name somewhere in the Title, Abstract or Key words. Or mention that it is weevil species you are dealing with.
The axis-legends and other texts in figs 2 and 3 (including the *) are so small that they are nearly impossible to read. Please use larger fonts. Fig. 2: no need to use both a symbol and a number, better use different symbols for species/sexes.
L. 155: there are no lowercase letters above the bars. Please add!
Fig. 2: Change the order of the two plots. In all other presentations, spring is mentions before summer.
Table 1, footnote: uppercase letter applies to species comparison for sexes separately?
l. 244: strobi
Author Response
Thank you for your time and work on our manuscript “Cuticular hydrocarbon recognition in the mating behavior of two Pissodes species”
The comments you provided have been fully considered and carefully treated as follow:
Q1: It appears (I.69-73) that the spring and the summer populations of the same year are two different generations, but we are not informed whether they have one or two generations per year. The question is relevant because the mating tests were made only with the summer population. If there is only one generation that breeds in spring (which is the typical life cycle where I come from), then the animals seem to mate both in summer and again (after hibernation) in spring, but the spring matings are probably the most important. In this sense I can understand why the author expect similar CHC composition and quantity at the two seasons. If there are two generations per year, this is not obvious. Please clear this up, and explain why only the summer animals were used for the mating test.
R1: This is a good question.
First, based on previous studies, these two sibling species only have one generation per year. In early spring, adults of both species emerge from their overwintering sites in the forest floor, feed, mate and lay eggs on their own habits. After hatching, the new adults emerge in late summer in the same year, feed for a while, disperse and go into hibernation until next spring. We reorganized Line 41-49 to be more precise.
These experiments were conducted to explore the CHC composition and function in mating behavior. There are many factors that influence insects mating choice, including age, mating experience and food sources. The spring adults can only be caught in the filed which contains too much uncertainty. The summer adults were newly emerged and collected in lab as we described in “materials and methods”. Compared to the spring adults, the condition of laboratory emerged adults are more consistent and suitable for this mating test.
Q2: Please add the family name somewhere in the Title, Abstract or Key words. Or mention that it is weevil species you are dealing with.
R2: The family name has been added in both abstract and introduction.
Q3: The axis-legends and other texts in figs 2 and 3 (including the *) are so small that they are nearly impossible o read. Please use larger fonts. Fig. 2: no need to use both a symbol and a number, better use different symbols for species/sexes.
R3: The fonts of the legends and * have been adjusted. Hope it is more readable now. The numbers were automatically formed during analysis. Symbols were applied by us later just to make the picture more readable.
Q4: There are no lowercase letters above the bars. Please add!
R4:The lowercase letters have been added.
Q5: Change the order of the two plots. In all other presentations, spring is mentions before summer.
R5: The order has been changed.
Q6: Table 1, footnote: uppercase letter applies to species comparison for sexes separately?
R6: Yes. The footnote has been rephrased.
Q7: L. 244 strobi
R7: The spelling has been corrected.
Again, thank you for all the work you’ve done on our manuscript. We are looking forward to your reply.
Reviewer 2 Report
Some suggestions are provided in the attached file. Excellent study.

Author Response
We greatly thankful for your time and work. The corrections we made based on your comments are in highlights.
1, P2, L41. The family name has been added.
2, P2, L61-62. The “I” has been changed into “i”.
3, P3, L92. The “p” has been changed into “P”.
4, P3, L119. spelling has been corrected.
5, P4, L135. We thought “step” would be a better word for this mating progress. “categories” might be suitable for parallel relationship.
6, P4, L145. The title has been modified.
7, P7, L228. The “p” has been changed into “P”.
8, P7, L253. The family name has been added.
9, P8, L259. The family name has been added.
Thank you for all the work you’ve done on our manuscript. We are looking forward to your reply.
Reviewer 3 Report
MDPI – Insects
Cuticular hydrocarbon recognition in the mating behavior of two Pissodes species
The authors of the submitted manuscript investigated the role of CHCs in mediating mating behaviour in two closely related weevil species. Presented results are interesting and provide evidence to explain why hybridisation occurs between the two species at certain points in the year. The manuscript is very will written with few spelling or grammatical errors.
Abstract and keywords
A well written abstract that provides the relevant information succinctly and accurately. The key words are appropriate. There are just two minor points to consider for the abstract:
· P1, L15: consider adding ‘weevil’ between ‘…sibling species…’
· P1, L23-24: clarify that ‘no significant differences’ refers to P. nemorensis
Introduction
The introduction provides a thoughtful, succinct summary of the existing knowledge surrounding the chemical ecology of these two weevil species and highlights key areas where further research is required. The authors have clearly stated the hypotheses they are testing and how they will be tested. Minor points to consider:
· P2, L40: add species describer information – e.g. P. strobe Peck and P. nemorensis Hopkins
· P2, L41: clarify that you are referring to the two weevil species and not the tree species
· P2, L42: this might be a personal preference, but try to avoid starting sentences with abbreviations
· P2, L48-49: very interesting that P. strobe produce the two pheromone components but don’t show any behavioural response to them
· P2, L57-63: inconsistent capitalisation of the numbered points
Materials and methods
The materials and methods would benefit from further detail in parts. Can the statistical analyses be done in software other than Excel? Key areas for improvement include:
· P2, L66-70: more detail could be provided for these locations – e.g. GPS coordinates etc.
· P2, L73-74: what temperature was the environmental chamber?
· P3, L80: what were the modifications?
· P3, L83: I am unclear as to what the 0.05 refers to, is it ml?
· P3, L98: hexane purity?
· P3, L104: what was the scan range of the MS?
· P3, L106: He flow rate?
· P3, L110: Kovats retention indices are not appropriate for a variable oven temperature programme as they are designed for isothermal oven temperature programmes. It would be more accurate and robust to use linear retention indices.
Results
The results are generally well presented and clear to understand. Just two minor points:
· P5, L155-156: there are no lowercase letters above the bars
· P6, L180-182: the asterisks would benefit from being larger, they are difficult to read
Discussion and conclusions
Well written, thoughtful discussion that places the results of the present study within the context of the known literature.
· P7, L239: replace ‘also existed’ with ‘were also present’
· P8, L265-266: I don’t think you provide any evidence to support the conclusion that CHC composition is influenced by food or environment
Author Response
We greatly thankful for your time and work. The corrections we made are in highlights. The main comments you provided in context have been fully considered and carefully treated as follow:
Q1: P1, L15: consider adding “weevil” between “… sibling speies…”
R1: P1, L15. The word has been added.
Q2: P1 L23-24: clarify that “no significant differences” refers to P. nemorensis
R2: P1, L25-26. This sentence has been rephrased and clarified that.
Q3: P2, L40: add species describer information – e.g. P. strobe Peck and P. nemorensis Hopkins
R3: P1, L15 and P1, L41. The information has been added.
Q4: P2, L41: clarify that you are referring to the two weevil species and not the tree species
R4: P1, L43. The sentence has been corrected to be more precise
Q5: P2, L42: this might be a personal preference, but try to avoid starting sentences with abbreviations
R5: P1, L45. We replaced the abbreviation with its corresponding full name.
Q6: P2, L48-49: Very interesting that P. strobi produce the two pheromone components but don’t show any behavioural response to them.
R5: Yes, it’s interesting. According to previous study, P. strobi only showed antennal electrophysiological respond to the two components but didn’t show any behavioral respond (Phillips and Lanier, 1986; Hibbard and Webster, 1993).
Q7: P2, L57-63: inconsistent capitalisation of the numbered points
R7: P2, L59-67. The numbered points have been consistent into number.
Q8: P2 L66-70: more detail could be provided for these locations -e.g. GPS coordinates etc.
R8: P2 L71-73. The detailed GPS locations have been provided.
Q9: P2 L73-74: what temperature was the environment chamber?
R9: P2 L78-80. We set a multiple temperature program. The program has been added to the manuscript.
Q10: P3, L80: What are the modifications?
R10: The modifications were made based on our laboratory conditions and the weevils. For example, the changes of Petri dish size, the amount of solvent used for CHC extraction, the extraction time, the amount of recoated FE, the mating test period et al. These details have been listed in “materials and methods”.
Q11: P3, L83: I am unclear as to what the 0.05 refers to, is it ml?
R11: P3, L89. The FE is the abbreviation of female of “female-equivalent”. 1 FE represent all the compounds that been extracted from a single female weevil.
Q12: P3, L98: hexane purity?
R12: P3, L107: The solvent details have been added to the manuscript.
Q13: P3, L104: what was the scan range of the MS?
R13; P3, L114. The MS scan rang was 33 – 650 m/z.
Q14: P3, L106: He flow rate?
R14: P3, L112-113. The He flow rate was 1 mL/min
Q15: P3, L110: Kovats retention indices are not appropriated for a variable oven temperature programme as they are designed for isothermal oven temperature programmes. It would be more accurate and robust to use linear retentions indices.
R15: P3, L116-L117. Thank you for point this out. The data we provided in the manuscript is calculated by linear retentions indices. We use a wrong word here. The mistake has been corrected.
Q16: P5, L155-156: there are no lowercase letters above the bars
R16: P5, L159. The no lowercase letters have been added above the bars
Q17: P6, L180-182: the asterisks would benefit from being larger, they are difficult to read
R17: The picture has been modified. Hope it is more readable now.
Q18: P7, L239: replace ‘also existed’ with ‘were also present
Q18: P7, L246. The sentence has been modified according to your suggestion.
Q19: P8, L265-266: I don’t think you provide any evidence to support the conclusion that CHC composition is influenced by food or environment.
R19: In spring, when the two species occupied different habits. The amount of CHC on P. strobi adults in spring is significantly more than P. nemorensis. Meanwhile, the amount of CHC on P. strobi adults in spring is also significantly more that in summer. We think the content variation could be an evidence to prove that CHC composition is influenced by the environment.
Thank you for all the work you’ve done on our manuscript. We are looking forward to your reply.